# Descriptive Sensory Attributes and Volatile Flavor Compounds of Plant-Based Meat Alternatives and Ground Beef

**DOI:** 10.3390/molecules28073151

**Published:** 2023-03-31

**Authors:** Manuel Sebastian Hernandez, Dale R. Woerner, J. Chance Brooks, Jerrad F. Legako

**Affiliations:** Department of Animal and Food Sciences, Texas Tech University, Lubbock, TX 79409, USA

**Keywords:** beef, plant-based meat alternatives, flavor, volatile compounds, texture, gas chromatography/mass spectrometry, sensory analysis, meat analogue

## Abstract

The objective of this study was to characterize descriptive sensory attributes and volatile compounds among ground beef (GB) and plant-based meat alternatives (PBMA). The Beyond Burger, Impossible Burger, a third brand of PBMA, regular GB, and lean GB were collected from local and national chain grocery stores. Patties were formed and cooked on an enamel-lined cast iron skillet to an internal temperature of 71 °C. A trained descriptive sensory panel evaluated patties for 17 flavor attributes and 4 texture attributes. Volatile compounds were extracted using solid phase microextraction and analyzed via gas chromatography-mass spectrometry. Distinct differences in sensory and volatile profiles were elucidated (*p* < 0.05). PBMA possessed decreased beef flavor intensity and increased umami, nutty, smokey-charcoal, and musty/earthy flavor compared to GB. Sensory differences corresponded with pyrazine, furan, ketone, alcohol, and aldehyde concentration differences between products. These data support the conclusion that ground beef and PBMA possess different flavor and texture characteristics. Furthermore, the flavor of PBMA varied among available retail brands.

## 1. Introduction

Plant-based meat alternatives (PBMA) have existed in the protein market since the 1980s. However, these products were often marketed directly to vegetarians [1]. More recently, a new generation of PBMA entered the market, claiming to directly mirror the beef hamburger eating experience [1]. Consumers are becoming more aware and conscious of how their food choices influence their health and the environment [2]. Therefore, these modernized PBMA piqued the interest of consumers who are opting to replace meat in their diet or decrease their meat consumption. The companies producing these new PBMA market their products as having the same flavor, aroma, texture, and appearance of beef hamburgers. Advancements in food science have opened the door for newer PBMA to more closely mimic beef compared to their predecessors [3]. For example, Impossible Foods Inc. employs the use of soy leghemoglobin to imitate “bleeding” in burgers as well as increase meat aroma intensity [4]. Despite these advancements, the flavor and aroma of PBMA are often not satisfactory to consumers [1,5]. Plant proteins have a beany aroma, flavor, and aftertaste [3,6]. The beany flavor is derived from the enzymatic oxidation of unsaturated fatty acids [6,7]. Godschalk-Broers et al. [5] reported that consumers had a lower expectation of good taste for various PBMA compared to real beef.

Beef flavor is incredibly complex and is influenced by a variety of intrinsic and extrinsic factors. Moreover, beef flavor is not a single attribute, but rather a culmination of various flavor aromatics [8]. The complexity of beef flavor is attributed to the three major flavor development pathways: the Maillard reaction, lipid degradation, and thiamine degradation [8,9,10]. Due to the “beany, astringent, bitter” aromas and tastes of plant-based proteins, various flavoring systems and ingredients are utilized to mimic beef flavor [3,6]. Davis et al. [11] reported that both a soy and pea protein-based PBMA did not possess beef flavor comparable to various lean levels of GB using both a consumer panel and trained panel. This study, however, did not capture specific flavor aromatics or basic tastes that could attribute to overall beef flavor. Kaczmarska et al. [12] used free choice profiling to characterize the aroma, flavors, tastes, texture, appearance, and mouthfeel of various traditional and new generation PBMA as well as of various meat products. Both soy and pea protein PBMA contained beefy aroma descriptors; however, beefy was not included in the flavor descriptors. The volatile profile of PBMA was comprised of alcohols, furans, and ketones, whereas the volatile profile of meat was primarily aldehydes, alcohols, and ketones [12]. Godschalk-Broers et al. [5] also reported decreased meat flavor intensity and flavor liking in the Beyond Burger compared to a beef burger.

To the best of our knowledge, limited literature exists that compares the PBMA and GB available at retail in the United States. Therefore, the objective of this study was to characterize the differences in sensory and volatile flavor profiles between ground beef (GB) and plant-based meat alternatives (PBMA) available at retail in the United States.

## 2. Results

### 2.1. Descriptive Sensory Analysis

The results of the flavor attribute assessment are presented in Table 1. All 17 flavor attributes were different due to product type (*p* ≤ 0.031). Beef flavor identity was most intense in RGB, followed by LGB (*p* < 0.05). Both GB products had more intense beef flavor identities compared to all PBMA (*p* < 0.05). Among PBMA, GEN had the lowest beef flavor intensity (*p* < 0.05) compared to IMP and BEY, which were similar (*p* > 0.05). GEN patties were less brown and roasted than all other products (*p* < 0.05). Brown intensity was greater in RGB compared to BEY (*p* < 0.05). Moreover, RGB had a greater roasted intensity compared to BEY and IMP (*p* < 0.05). Fat-like was most intense in BEY compared to all other products (*p* < 0.05). Fat-like intensity was greater in IMP compared to both lean levels of GB (*p* < 0.05). As expected, fat-like intensity was greater in RGB compared to LGB (*p* < 0.05). GEN had the lowest fat-like intensity compared to all other products (*p* < 0.05). No bloody/serumy notes were detected in PBMA. In both GB products, bloody/serum notes were detected at very low intensities and were similar (*p* > 0.05). Buttery was the most intense in BEY and IMP patties compared to all other products (*p* < 0.05). Both GB products had similar buttery intensities (*p* > 0.05). GEN had the lowest buttery intensity compared to all other products (*p* < 0.05). Both lean levels of GB had lower overall sweet notes compared to PBMA (*p* < 0.05). Smokey-charcoal intensity was greater in PBMA compared to both GB products (*p* < 0.05). Moreover, GEN produced more smokey-charcoal notes than BEY (*p* < 0.05). Umami intensity was the lowest in both GB products compared to PBMA (*p* < 0.05). Among PBMA, BEY was most intense for umami, followed by IMP, then GEN (*p* < 0.05). Metallic, liver-like, and oxidized were the most intense in both GB products compared to PBMA (*p* < 0.05). Neither metallic nor oxidized were detected in IMP. LGB was more livery than RGB (*p* < 0.05). Nutty and musty/earthy notes were more intense in PBMA than in GB (*p* < 0.05). Among PBMA, BEY and IMP were nuttier than GEN (*p* < 0.05). Moreover, musty/earthy notes were more intense in GEN and IMP compared to BEY (*p* < 0.05). LGB, IMP, and GEN patties were more bitter than BEY (*p* < 0.05). All PBMA were saltier than both GB products (*p* < 0.05). Among PBMA, BEY were the saltiest, followed by IMP, then GEN (*p* < 0.05). Both lean levels of GB were more sour than BEY and IMP (*p* < 0.05). Sour was not detected in BEY.

The results of the texture attribute assessment are presented in Table 2. BEY patties were juicier than all other products (*p* < 0.05). Juiciness was similar between both GB products and IMP (*p* > 0.05). GEN patties were the driest compared to all other products (*p* < 0.05). IMP patties were more cohesive than LGB, BEY, and GEN (*p* < 0.05). Both GB products were harder than PBMA (*p* < 0.05). GEN patties were softer compared to all other products (*p* < 0.05). The particle sizes of masticated patties were similar among both lean levels of GB and BEY and IMP (*p* < 0.05). BEY patties had a larger particle size compared to IMP and GEN patties (*p* < 0.05).

### 2.2. Volatile Compound Analysis

The results of the analysis of Maillard reaction-derived volatile compounds are presented in Table 3. Furfural, 2-furanmethanol, and 5-methylfurfural production was increased in BEY and GEN patties (*p* < 0.05) compared to IMP and both lean levels of GB, which were similar (*p* > 0.05). IMP and GEN had greater concentrations of 2-methyl-3-furanthiol compared to RGB (*p* < 0.05). 2-Methyl-3-furanthiol concentrations were similar between LGB, BEY, and IMP (*p* < 0.05). 2-Methyltetrahydro-3-furanone concentration was increased in GEN compared to IMP (*p* < 0.05). RGB produced the greatest concentration of acetoin compared to all other products (*p* < 0.05). Moreover, acetoin concentration was greater in LGB compared to BEY and IMP (*p* < 0.05). 2,3-Butanedione concentration was greater in RGB compared to all PBMA (*p* < 0.05). LGB had a greater concentration of 2,3-butanedione compared to BEY and IMP (*p* < 0.05). Methylpyrazine and 2,5-dimethylpyrazine concentrations were greater in PBMA compared to both GB products (*p* < 0.05). Trimethylpyrazine concentrations were the greatest in GEN patties compared to all other products (*p* < 0.05). Moreover, both GEN and IMP patties produced greater concentrations of trimethylpyrazine compared to both GB products (*p* < 0.05). GEN patties produced the greatest concentration of 2-ethyl-3,5/6-dimethylpyrazine (*p* < 0.05) compared to all other products, which were similar (*p* > 0.05). BEY and GEN patties produced a greater concentration of 2-acetylpyrrole compared to both GB products (*p* < 0.05). Both GB products produced similar concentrations of 2-acetylpyrrole compared to IMP (*p* < 0.05). Of the Strecker aldehydes, acetaldehyde and 3-methylbutanal were not influenced by product type (*p* ≥ 0.179). IMP produced the greatest concentration of benzaldehyde, butyraldehyde, and phenylacetaldehyde compared to all other products (*p* < 0.05). Both GB products produced less benzaldehyde compared to GEN (*p* < 0.05). Methional concentration was the greatest in BEY compared to all other products (*p* < 0.05). RGB produced a greater concentration of phenylacetaldehyde compared to BEY (*p* < 0.05). Both GB products produced a greater concentration of 2-methylbutanal compared to BEY (*p* < 0.05). Moreover, RGB produced a greater concentration of 2-methylbutanal compared to GEN (*p* < 0.05). Of the sulfur-containing compounds, dimethyl sulfide, furfuryl sulfide, and methanethiol were not influenced by product type (*p* ≥ 0.158). IMP and LGB produced similar concentrations of carbon disulfide (*p* > 0.05). Carbon disulfide concentration was greater in LGB compared to BEY and GEN (*p* < 0.05). Diallyl sulfide concentration was the greatest in IMP and GEN compared to all other products (*p* < 0.05). Dimethyl disulfide concentration was greater in IMP compared to BEY and both GB products (*p* < 0.05). IMP produced the greatest concentration of 3-methyl thiophene compared to all other products (*p* < 0.05).

The results of the lipid-derived volatile compound analysis are reported in Table 4. Of the alcohols, 1-hexanol and 2,3-butanediol were not influenced by product type (*p* ≥ 0.052). Ethanol concentration was the greatest in GEN compared to all other products (*p* < 0.05). IMP produced the greatest concentration of 1-octanol compared to all other products (*p* < 0.05). RGB produced the greatest concentration of 1-octen-3-ol and 1-pentanol (*p* < 0.05). 1-Penten-3-ol concentration was greater in both GB products compared to BEY (*p* < 0.05). Both GB products had similar 1-penten-3-ol concentrations as IMP for (*p* > 0.05). Moreover, IMP produced a greater concentration of 1-penten-3-ol compared to BEY and GEN (*p* < 0.05). Decanal and dodecanal were not influenced by product type (*p* ≥ 0.106). LGB produced lower concentrations of heptanal compared to RGB and BEY (*p* < 0.05) but was similar to IMP and GEN (*p* > 0.05). Both lean levels of GB produced less hexanal compared to BEY (*p* < 0.05). Nonanal and pentanal concentrations were greater in BEY and GEN compared to LGB (*p* < 0.05). Octanal concentration was similar between RGB and all PBMA (*p* > 0.05). LGB had a lower concentration of octanal compared to BEY (*p* < 0.05). GEN patties produced the greatest concentration of 2-undecenal compared to all other products (*p* < 0.05). BEY produced a greater concentration of 2,4-decadienal compared to IMP and both lean levels of GB (*p* < 0.05). Toluene was not influenced by product type (*p* = 0.364). d-Limonene and α-pinene concentrations were the greatest in GEN compared to all other products (*p* < 0.05). LGB produced a greater concentration of styrene compared to IMP and GEN (*p* < 0.05). BEY produced a greater concentration of styrene compared to RGB (*p* < 0.05). Both BEY and IMP produced the greatest concentration of *p*-xylene compared to GEN and both GB products (*p* < 0.05). Heptanoic acid and nonanoic acid production were not influenced by product type (*p* ≥ 0.207). Both GB products produced less acetic acid compared to IMP and GEN (*p* < 0.05). Butanoic acid concentration was greater in RGB compared to IMP (*p* < 0.05). GEN produced the greatest concentration of octanoic acid compared to all other products (*p* < 0.05). IMP produced the greatest concentration of methyl butyrate compared to all other products (*p* < 0.05). Methyl hexanoate concentration was greater in RGB compared to IMP (*p* < 0.05). Methyl nonanoate concentration was the greatest in GEN compared to all other products (*p* < 0.05). GEN produced a greater concentration of methyl octanoate compared to RGB (*p* < 0.05). RGB produced a greater concentration of methyl propionate compared to BEY and GEN (*p* < 0.05). BEY and GEN produced the greatest concentration of 2-pentyl furan compared to all other products (*p* < 0.05). Benzene and decane production were not influenced by product type (*p* ≥ 0.244). LGB produced a greater concentration of ethylbenzene compared to IMP and GEN (*p* < 0.05). BEY had a greater concentration of nonane compared to all other products (*p* < 0.05). Moreover, RGB produced a greater concentration of nonane compared to IMP (*p* < 0.05). RGB and BEY produced the greatest concentration of octane compared to all other products (*p* < 0.05). 1-Octene concentration was the greatest in RGB compared to all other products (*p* < 0.05). Butyrolactone and 2-pentanone concentration was not influenced by product (*p* < 0.05). IMP produced the greatest concentrations of 2-butanone and 2-propanone compared to all other products (*p* < 0.05). 2-Heptanone concentration was the greatest in BEY compared to all other products (*p* < 0.05).

Agglomerative hierarchical clustering was conducted to visualize similarities and differences between the total volatile flavor profiles between PBMA and GB (Figure 1). Two clusters were generated via AHC, where the BEY and GEN were clustered together and kept separate from IMP and both lean levels of GB. Strecker aldehydes seemed to be the cause of IMP clustering with both GB products. However, within the IMP and GB cluster, both GB products clustered separate from IMP.

### 2.3. Relationship between Volatile Compounds and Sensory Attributes

To understand the influence of volatile flavor development on descriptive flavor attributes, partial least squares regression (PLSR) was conducted (Figure 2). Flavor attributes served as the dependent variables, and volatile compounds served as the explanatory variables. The T1 axis separated PBMA from GB. Both GB products were primarily associated with beef flavor, bloody/serum, metallic, oxidized, and liver-like aromatics as well as sour basic taste. The volatiles primarily associated with GB and these flavor aromatics were 2-methylbutanal, acetaldehyde, acetoin, 2,3-butanedione, butyrolactone, butanoic acid, 1-penten-3-ol, 1-octen-3-ol, methyl propionate, and methyl hexanoate. Other flavor attributes that loaded positively on axis T1 were brown and roasted aromatics and sour basic taste. 2-Pentanone and toluene were associated with brown and roasted. All three PBMA negatively loaded on axis T1 and were associated with musty/earthy, smokey-charcoal, overall sweet, nutty, buttery, and fat-like aromatics as well as salty and umami basic tastes. Volatile compounds associated with these flavor attributes included furans, pyrazines, ketones, aldehydes, carboxylic acids, esters, oxolanes, and sulfur-containing compounds. 

## 3. Discussion

The ultimate goal of PBMA is to produce a product that directly imitates ground beef. Kaczmarska, et al. [12] reported plant-based protein and meat substitutes produced a greater diversity of volatile compounds compared to beef and other animal protein sources, which translated to the difference in sensory descriptors. The present study shows PBMA do not possess characteristic beef flavor aromatics; rather, they comprised other flavor aromatics that are not always detectable in beef, such as nutty, smokey-charcoal, and musty/earthy [13,14]. These results are congruent with the findings of Davis et al. [11], who reported a clear difference in beef flavor identity between GB and PBMA. Beef flavor is derived from the Maillard reaction, lipid degradation, their interaction, and thiamine degradation [8,10]. The Maillard reaction is responsible for the development of meaty, roasted, and grilled aromas [10]. However, lipid degradation is responsible for meat animal species-specific flavor [10]. It is clear that the lipids present in PBMA did not degrade into flavor volatiles that produce beef flavor aromatics. This is to be expected, as PBMA are formulated with various vegetable and plant oils. For instance, the Beyond Burger uses refined coconut oil and expeller-pressed canola oil as lipid sources. He et al. [15] reported that the fatty acid profiles of GB and PBMA were distinct, with PBMA comprising a greater percentage of mono- and polyunsaturated fatty acids. Butanoic acid and 1-octen-3-ol were reported to contribute to characteristic beef flavor, and this is supported by the present study, where RGB had the most intense beef flavor and the greatest concentration of butanoic acid and 1-octen-3-ol [16,17]. Moreover, the PLSR biplot (Figure 2) shows both 1-octen-3-ol and butanoic acid clustering with beef flavor identity and both GB products. Other lipid-derived compounds reported to contribute to beef flavor are hexanal, (E,E)-2,4-decadienal, (E,E)-2,4-heptadienal, and octanoic acid [16]. It should be noted that these conclusions were in heated beef tallow [16] and fractionated beef fat [17], subsequently removing Maillard and other water-soluble precursor interactions. All aforementioned volatiles were also reported to contribute to off-odors and flavors if produced from lipid autooxidation [8,10,18]. However, thermal lipid degradation produces lipid-derived compounds that possess desirable aromas [8,18]. The interactions between lipid- and Maillard reaction-derived compounds were also suggested to be very important for the development of characteristic beef flavor [8,18]. It could be speculated that the lipid-derived compounds from plant lipid sources interact with Maillard reaction products to produce aromatics that are “meaty” but not inherently “beefy”. The objective of this study was ultimately to compare the flavor profile of PBMA to ground beef. However, this study could have benefitted from capturing sensory data for an overall “meaty” aroma to further elucidate the flavor profile of PBMA.

Generally, PBMA produced greater concentrations of Maillard reaction products. These volatile compounds contributed to the development of nutty and smokey-charcoal aromatics. Various flavoring systems and/or ingredients are readily used in PBMA to produce a meaty aroma [3,6]. A common flavor system in PBMA utilizes cysteine and ribose Maillard systems [6]. These flavor precursors were suggested to be potent substrates for the Maillard reaction and the development of meat flavor [10]. However, ribose is expensive; therefore, its isomer, xylose, is used. This potential increase in xylose may explain the increase of Maillard compounds contributing to aromatics not associated with beef flavor. In the current study, PBMA showed a dramatic increase in furfural concentration. The aroma descriptors of furfural include sweet, brown, nutty, woody, bready, and caramellic [8]. This would suggest that the Maillard reaction products produced by flavor systems are useful for the development of browned and even “meaty” aromas but not characteristic beef aroma. In the present study, two Maillard-derived ketones, 2,3-butanedione and acetoin, were present in greater concentrations in GB compared to PBMA. This result is in contrast with the findings of Kaczmarska et al. [12], who reported 2,3-butanedione concentration was greater in PBMA compared to beef. These compounds are normally associated with butter aroma and flavor. Despite these compounds having greater concentrations in GB, two of the PBMA products (BEY and GEN) had a more intense buttery aromatic. This may suggest that the 2,3-butanedione and acetoin concentrations reported in the current study may be optimal for the development of characteristic beef flavor. The increase of Maillard reaction products in PBMA agrees with He et al. [15], who reported an increase of pyrazines, Strecker aldehydes, and sulfur-containing compounds in new-generation PBMA compared to GB. The authors attributed this increase to the flavorings and other ingredients present in PBMA. Additionally, it was suggested that the production processes of PBMA contributed to the development of Maillard reaction products, as Maillard-derived compounds were present in raw PBMA [15]. The extrusion of plant proteins generates flavor precursors via hydrolysis, which, under high-temperature processing, undergo the Maillard reaction [7,19].

Sulfur-containing compounds (aldehydes, sulfides, thiols, and thiophenes) are important contributors to meat flavor and aroma [9,10]. However, sulfur-containing compounds are also primary contributors of vegetable and mushroom flavors and aromas [20]. This abundance of sulfur compounds is due to the various precursors (sulfur-containing amino acids, thiamine, and glucosinolates) found in both animal and plant food products. Regarding meat flavor development, cysteine/ribose-derived sulfur compounds are suggested to be important for characteristic meat flavor [10]. Moreover, the Strecker degradation of cysteine results in hydrogen sulfide, which can react with other Maillard intermediates to produce more classes of volatile compounds [10]. The production of sulfur-containing compounds can be mediated by the presence of phospholipids. Both Farmer et al. [21] and Farmer and Mottram [22] reported that various animal-derived phospholipids and beef triglycerides present in a cysteine/ribose Maillard reaction system reduced inappropriate sulfurous aromas and increased meaty aroma. As previously discussed, lipid–Maillard interactions are important for the development of beef flavor. Generally, IMP and GEN patties produced greater concentrations of sulfur-containing compounds. The lipid profiles, i.e., fatty acids, of PBMA are distinct from GB [15]; therefore, it is likely that the phospholipid profiles of PBMA and GB are different and may contribute to the differences in sulfur-containing volatile compound concentrations. Furthermore, the addition of thiamin to PBMA may also explain the increase in sulfur-containing compounds. Diallyl sulfide is responsible for garlic aromas and is derived from *Brassica* and Allium vegetables [20]. However, it can appear in animal products as well. In the present study, GEN and IMP patties had an approximately 20% increase in diallyl sulfide compared to GB. This increase would also explain the deviation of PBMA flavor away from characteristic beef flavor.

Other ingredients are used in PBMA to generate meaty aromas. Yeast extract is readily used in PBMA to produce “meaty” aromas due to its composition of meat flavor precursors, i.e., reducing sugars, amino acids, peptides, and thiamine [23]. Pyrazines and thiophenes are endogenous to yeast and produce a sweet, roasted meat aroma [6]. Moreover, when exposed to heat, yeast extract produces a variety of volatile compounds that contribute to meaty aromas [23]. Yeast extract and dried yeast are present in the ingredient statements of the PBMA used in the current study; therefore, this could explain the increase in some Maillard reaction products in PBMA. Another ingredient that was marketed by a PBMA company to develop meat flavor is heme iron. The role of heme iron in positive meat flavor development has not been fully elucidated in the literature. However, Devaere et al. [24] reported an increase in hexanal, pyrazines, and furans and a decrease in Strecker aldehydes in a PBMA formula supplemented with commercial bovine myoglobin (0.5% and 1.0%). This increase in hexanal and pyrazines in the PBMA that uses heme iron (IMP) corroborates the results in Devaere et al. [24]. Moreover, the reduction of Strecker aldehydes in IMP compared to BEY and GEN could potentially be the result of the heme iron. The heme iron in myoglobin is suggested to serve as a cofactor when exposed during thermal denaturation of myoglobin, i.e., cooking, and catalyzes meat flavor precursor reactions [4]. The heme iron in PBMA also serves to portray the serum/metallic notes in rare beef. However, this was not observed by the descriptive sensory panel in the current study. In contrast, there are more reports of heme iron being a contributor to off-flavor development, as heme iron serves as a pro-oxidant, resulting in oxidized and liver flavor aromatics. Moreover, there are no available data indicating that soy leghemoglobin, the plant derivative of heme iron, results in meat flavor development.

Plant proteins are often associated with beany, astringent, and bitter attributes [3,6,7]. These attributes are a result of the oxidation of unsaturated fatty acids as well as taste-active plant metabolites such as saponins, isoflavones, and phytosterols [3,7]. However, descriptive sensory analysis in the present study showed few differences in bitter basic taste. As previously mentioned, attributes such as bitter are masked through the inclusion of flavor systems and/or ingredients [6]. The present study showed an increase in umami and salty intensities in PBMA. This increase could be derived from the added salt and yeast extract. As previously discussed, yeast extract is rich in taste-active compounds that can contribute to umami and salty tastes, i.e., amino acids and peptides [23]. Additionally, Swing et al. [25] reported an increased concentration of glutamic acid in PBMA compared to ground pork and the USDA Nutrient Database value of GB. This increase in glutamic acid would explain the increase in umami intensity [26]. Kaczmarska et al. [12] used free choice profiling to measure flavor, aroma, texture, and mouthfeel in plant-based meat alternatives and plant-based proteins such as tofu and tempeh. The PBMA flavor and aroma descriptors included “meaty, salty, mushroom, umami/MSG, smokey” and coincided with the ingredients found in the products [12]. Although a qualitative approach was used in the aforementioned study, these descriptors are similar to the attributes evaluated in the present study and were generally more intense in PBMA compared to GB.

Both differences and similarities were observed in texture attributes. Godschalk-Broers et al. [5] reported untrained consumer panelists rated a beef burger as harder, more cohesive, and juicier than the Beyond Burger. The present study is in partial agreement with the aforementioned study, as both GB products were rated harder than the PBMA. Davis et al. [11] evaluated the texture of PBMA using texture profile analysis. In the present study, GB was harder than all PBMA, which is congruent with the instrumental hardness reported in [11]. Moreover, GB was more cohesive than BEY, supporting the instrumental cohesiveness results presented by Davis et al. [11]. However, in the present study, IMP had similar cohesiveness compared to RGB, which deviates from the findings of Davis et al. [11]. Moreover, differences in juiciness between GB and PBMA were not observed in the present study, unlike those reported in Godschalk-Broers et al. [5] and Davis et al. [11]. These discrepancies could be explained by changes in the formulation and/or cooking method (cast-iron skillet vs. clamshell electric grill). Overall, the texture intensities of both GB and PBMA were relative to the intensities of 80/20 ground beef reported in Miller et al. [14].

## 4. Materials and Methods

### 4.1. Product Collection and Patty Manufacturing

Plant-based meat alternatives including the Beyond Burger (BEY), Impossible Burger (IMP), and a third available brand of plant-based protein (GEN) as well as regular ground beef (RGB; 80 to 85% lean) and lean ground beef (LGB; ≥93% lean) were collected from local and national chain grocery stores in six different cities representing the east (University Park, PA, USA and Athens, GA, USA), central (West Lafayette, IN, USA and Lubbock, TX, USA), and west (Fresno, CA, USA and Reno, NV, USA) regions of the United States. In each city, 6 packages of each product type were purchased from at least 2 different stores. After collection, product was shipped overnight to Texas Tech University with refrigerant materials. Missing or compromised product was replaced with product purchased in Lubbock, TX. If 85/15 GB was not available, 80/20 GB was purchased. If 93/7 GB was not available, 96/4 or 97/3 GB was purchased. All product was collected in the same week. One-hundred and fifty grams of product were weighed out and formed into a patty using a patty press (Gander Mountain, St. Paul, MN, USA). From 1 package, 2 patties were produced and assigned to either volatile compound analysis or sensory analysis. Patties were vacuum-packaged and frozen at −20 °C until subsequent analyses.

### 4.2. Cooking Procedure

Prior to cooking, patties were thawed for 24 h at 0 to 4 °C. Patties were cooked on an enamel-lined cast-iron skillet (80131, Tramontina USA Inc., Sugar Land, TX, USA) heated to a surface temperature of 200 ± 10 °C. Skillet surface temperature was monitored using a laser infrared thermometer (Model 66IR30, Amazon Commercial, Bellevue, WA, USA). Patties were cooked to an internal temperature of 71 °C and were flipped at 35 °C. Raw weight, cooked weight, and peak temperature were recorded.

### 4.3. Descriptive Sensory Analysis

Descriptive sensory analysis was conducted according to the American Meat Science Association Sensory Guidelines [27]. Panelists (*n* = 9) were trained to identify and quantify the intensity of 21 flavor and texture attributes from Adhikari et al. [10] and AMSA [27], which are reported in Appendix A. Attributes were rated on a 100-point scale, where 0 = not detectable and 100 = extremely intense. Panelists were trained for 20 h prior to testing. Patties for descriptive sensory evaluation were cooked as previously described. Following cooking, steaks were wrapped in aluminum foil, held at 50 to 55 °C in a food service warmer (Cambro Manufacturing, Huntington Beach, CA, USA), and served within 10 min. Patties were cut into 6 wedges and served in souffle cups. Panelists evaluated 1 wedge under red gel lights and recorded attribute scores using a digital survey on a tablet (Qualtrics Surveys, Provo, UT, USA; iPad, Apple Inc., Cupertino, CA, USA). Prior to the first sample and in between samples, panelists were instructed to cleanse their palate with apple juice, saltless crackers, and distilled water. Panelists were also provided an expectorant cup, napkin, and toothpick. Ten samples, in random order, were evaluated per session with a 10 min break between the 5th and 6th sample. Each product type was evaluated twice in 1 panel session.

### 4.4. Volatile Compound Analysis

Volatile compounds were determined using the modified methods of Hernandez et al. [28]. Patties designated for volatile compound analysis were cooked as previously described. Immediately following cooking, patties were cut into cubes, flash-frozen with liquid nitrogen, and homogenized (Robot Coupe, Blixer 3 Food Processor, Robot Coupe, Jackson, MS). Five grams of cooked homogenate were weighed into 20 mL glass vials and spiked with 10 µL of an internal standard solution (1,2 dichlorobenzene, 2.5 µg/ µL). Vials were sealed with a 1.3-mm polytetrafluoroethylene septa and metal screw cap (Gerstel Inc., Linthicum, MD, USA). Samples were loaded into a dry air-cooling block set to −5 °C (MeCour Temperature Control, LLC, Groveland, MA, USA). An autosampler (Multipurpose Sampler, Gerstel, Inc., Linthicum, MD, USA) removed samples from the cooling block and placed them in an agitator for a 5-min incubation period at 65 °C. Following incubation, a 25-min extraction period was used to collect volatile compounds from the sample headspace via solid phase microextraction (SPME) with an 85 µm film thickness carboxen polydimethylsiloxane fiber (Supelco Inc., Bellefonte, PA, USA). After extraction, the SPME fiber was injected into the GC (7890B series, Agilent, Santa Clara, CA, USA) and desorbed onto a VF-5ms capillary column (30 m × 0.25 mm × 1 µm; Agilent J&W GC columns, Santa Clara, CA, USA). Column eluates were introduced into the single quadrupole mass spectrometer (5977A, Agilent, Santa Clara, CA, USA) via electron ionization at 70 eV. Volatile compounds were detected within a mass range of 45–500 *m/z*. Data were acquired in selective ion monitoring and full scan modes. External analytical grade standards (Sigma-Aldrich, St. Louis, MO, USA) were used to confirm compound identities through retention time and the fragmentation patterns of 3 key ions (Appendix A). Quantitation of volatile compounds (ng per gram of sample) was conducted using the internal standard and a 5-level calibration curve.

### 4.5. Statistical Analysis

Sensory and volatile data were analyzed as a randomized complete block design, with 6 blocks representing collection city and 6 replications per product type. Product package served as the experimental unit. Statistical analyses were conducted using the PROC GLIMMIX procedure of SAS (v. 9.4, Cary, NC, USA), where product served as a fixed effect and collection city served as a random effect. For sensory data, panel session and feed order were included as random effects. When significant (*p* < 0.05), patty peak temperature was included as a covariate. The Kenward–Roger adjustment was used to estimate denominator degrees of freedom. Least squares means were separated using the PDIFF function. An alpha of 0.05 was used for all analyses. Agglomerative hierarchical clustering was utilized to understand how product type influenced volatile flavor profiles. To understand relationships between sensory and volatile data, partial least squares regression was conducted using XLSTAT (v 2022.1, Addinsoft, Paris, France), with volatile compounds serving as the Y factor and descriptive flavor attributes serving as the X factor. Data are presented in a biplot.

## 5. Conclusions

This study is one of the first to characterize the sensory and volatile differences between PBMA and GB. Moreover, this study included PBMA formulated with various plant proteins. PBMA had lower beef flavor intensity compared to GB, suggesting the meaty aroma of PBMA is not comparable to beef flavor. PBMA produced greater concentrations of Maillard reaction products. Of the PBMA, the volatile flavor profile of IMP was most similar to GB regarding Strecker aldehydes. The flavor profiles of PBMA and GB are very complex. Regarding PBMA, flavor profiles are derived from flavor systems, added ingredients, and production processes. As for GB, characteristic beef flavor is derived from the Maillard reaction, thermal lipid degradation, and their interactions. These data further support that the flavor dynamics of foods are very complex and can be altered through various factors; thus, no single factor can be the sole contributor to overall flavor profile. Overall, the sensory and chemical flavor profiles of PBMA and GB differ.

## Figures and Tables

**Figure 1 molecules-28-03151-f001:**
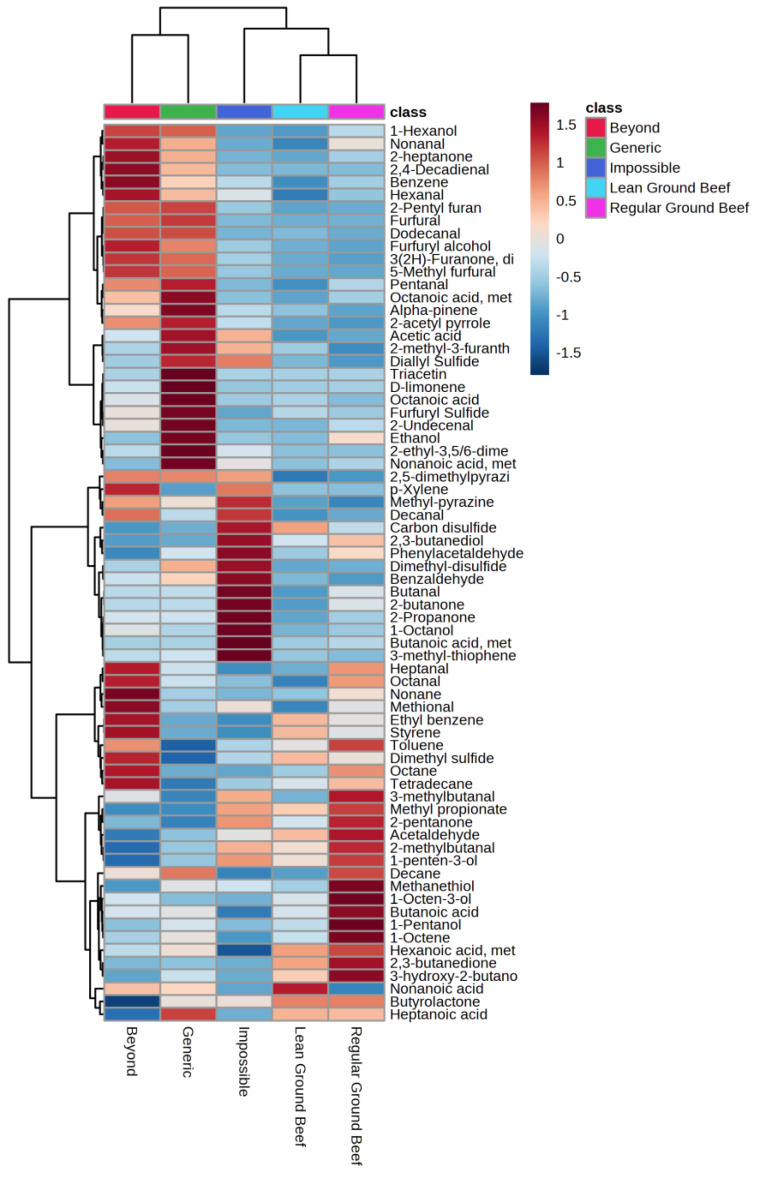
Agglomerative hierarchical clustering of quantitated volatile compound concentration (ng/g) from ground beef and plant-based meat alternatives.

**Figure 2 molecules-28-03151-f002:**
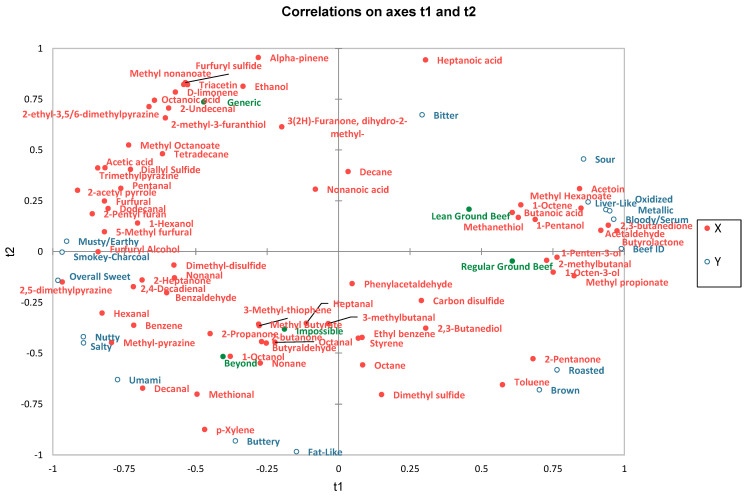
Partial least squares regression bi-plot of ground beef and plant-based meat alternatives where descriptive flavor attribute served was the dependent variable and quantitated volatile compound concentration (ng/g) served as the explanatory variable.

**Table 1 molecules-28-03151-t001:** Least squares means of descriptive flavor attributes ^1^ from three plant-based meat alternatives and two lean levels of ground beef.

	Ground Beef		Plant-Based Meat Alternatives				
Attribute	Lean ^2^	Regular ^3^	Beyond Meat	Impossible Burger	Third Retail Brand	SEM ^4^	*p*-Value ^5^
Beef Flavor Identity	46.8 ^b^	51.0 ^a^	16.7 ^c^	17.5 ^c^	11.5 ^d^	1.10	<0.001
Brown	49.2 ^ab^	50.2 ^a^	48.3 ^b^	49.6 ^ab^	45.4 ^c^	0.67	<0.001
Roasted	49.7 ^ab^	50.6 ^a^	49.0 ^b^	47.6 ^c^	44.0 ^d^	0.43	<0.001
Fat-Like	15.8 ^d^	17.2 ^c^	19.6 ^a^	18.5 ^b^	14.5 ^e^	0.48	<0.001
Bloody/Serum	0.8 ^a^	0.7 ^a^	0.0 ^b^	0.0 ^b^	0.0 ^b^	0.13	<0.001
Buttery	1.4 ^bc^	2.0 ^b^	4.2 ^a^	3.6 ^a^	1.0 ^c^	0.28	<0.001
Overall Sweet	2.1 ^b^	1.1 ^b^	13.1 ^a^	12.4 ^a^	12.7 ^a^	0.44	<0.001
Smokey-Charcoal	4.1 ^c^	3.3 ^c^	9.6 ^b^	10.6 ^ab^	11.1 ^a^	0.73	<0.001
Umami	14.4 ^d^	13.9 ^d^	24.9 ^a^	22.4 ^b^	17.5 ^c^	0.46	<0.001
Metallic	1.5 ^a^	1.1 ^a^	0.1 ^b^	0.0 ^b^	0.1 ^b^	0.17	<0.001
Liver-Like	1.5 ^a^	0.9 ^b^	0.1 ^c^	0.0 ^c^	0.2 ^c^	0.25	<0.001
Oxidized	3.7 ^a^	3.3 ^a^	0.2 ^b^	0.1 ^b^	0.5 ^b^	0.38	<0.001
Nutty	0.8 ^c^	0.0 ^c^	12.1 ^a^	11.8 ^a^	7.7 ^b^	0.51	<0.001
Musty/Earthy	2.8 ^c^	2.3 ^c^	9.4 ^b^	11.8 ^a^	12.2 ^a^	0.60	<0.001
Bitter	3.6 ^a^	2.9 ^ab^	1.8 ^b^	3.4 ^a^	3.5 ^a^	0.65	0.031
Salty	1.7 ^d^	0.7 ^d^	15.0 ^a^	12.1 ^b^	8.7 ^c^	0.48	<0.001
Sour	0.6 ^a^	0.5 ^ab^	0.0 ^c^	0.1 ^c^	0.2 ^bc^	0.11	<0.001

^1^ Attributes from the Beef Flavor Lexicon Adhikari et al. [10]. ^2^ Includes 15% and 20% fat ground beef. ^3^ Includes less than or equal to 7% fat ground beef. ^4^ Largest standard error of the least squares means. ^5^ Observed significance level. ^a–e^ Means without a common superscript differ (*p* < 0.05).

**Table 2 molecules-28-03151-t002:** Least squares means of descriptive texture attributes ^1^ from three plant-based meat alternatives and two lean levels of ground beef.

	Ground Beef		Plant-Based Meat Alternatives				
Attribute	Lean ^2^	Regular ^3^	Beyond Meat	Impossible Burger	Third Retail Brand	SEM ^2^	*p*-Value ^3^
Juiciness	51.3 ^b^	52.1 ^b^	55.1 ^a^	51.3 ^b^	42.9 ^c^	0.74	<0.001
Cohesiveness	28.5 ^b^	29.4 ^ab^	24.5 ^c^	31.1 ^a^	28.5 ^b^	1.07	<0.001
Hardness	29.9 ^a^	30.6 ^a^	27.9 ^b^	27.6 ^b^	25.5 ^c^	0.89	<0.001
Particle Size	31.0 ^ab^	31.2 ^ab^	31.9 ^a^	30.1 ^b^	23.5 ^c^	0.58	<0.001

^1^ Attributes from the AMSA Sensory Guidelines AMSA. ^2^ Includes 15% and 20% fat ground beef. ^3^ Includes less than or equal to 7% fat ground beef. ^a–c^ Means without a common superscript differ (*p* < 0.05).

**Table 3 molecules-28-03151-t003:** Least squares means of Maillard reaction-derived volatile compound concentrations from three plant-based meat alternatives and two lean levels of ground beef.

	Ground Beef		Plant-Based Meat Alternatives				
Volatile Compound (ng/g)	Lean ^1^	Regular ^2^	Beyond Meat	Impossible Burger	Third Retail Brand	SEM ^3^	*p*-Value ^4^
*Furans*							
Furfural	18.67 ^b^	20.99 ^b^	987.41 ^a^	64.71 ^b^	1093.54 ^a^	138.580	<0.001
2-Furan methanol	23.19 ^b^	16.25 ^b^	175.73 ^a^	39.18 ^b^	136.85 ^a^	19.454	<0.001
2-Methyl-3-furanthiol	21.33 ^abc^	1.58 ^c^	11.11 ^bc^	24.04 ^ab^	41.09 ^a^	9.246	<0.001
5-Methyl furfural	6.24 ^b^	5.76 ^b^	24.33 ^a^	6.99 ^b^	21.74 ^a^	2.194	<0.001
*Ketones*							
Acetoin	13.12 ^b^	24.93 ^a^	2.27 ^c^	2.76 ^c^	8.48 ^bc^	3.915	<0.001
2,3-Butanedione	30.15 ^ab^	42.01 ^a^	12.79 ^c^	11.84 ^c^	13.95 ^bc^	6.348	0.002
*Oxolanes*							
2-Methyltetrahydro-3-furanone	15.08 ^ab^	12.02 ^ab^	14.50 ^ab^	9.70 ^b^	15.95 ^a^	4.111	0.005
*Pyrazines*							
Methylpyrazine	12.73 ^c^	10.26 ^c^	30.88 ^ab^	38.94 ^a^	24.18 ^b^	5.303	<0.001
Trimethylpyrazine	2.98 ^d^	3.91 ^cd^	7.82 ^bc^	9.67 ^b^	15.56 ^a^	1.707	<0.001
2,5-Dimethylpyrazine	14.46 ^b^	16.97 ^b^	32.37 ^a^	30.92 ^a^	32.09 ^a^	5.203	0.017
2-Ethyl-3,5/6-dimethylpyrazine	6.73 ^b^	7.35 ^b^	10.33 ^b^	12.48 ^b^	36.20 ^a^	6.338	0.001
*Pyrroles*							
2-Acetylpyrrole	169.69 ^c^	141.75 ^c^	559.78 ^ab^	306.20 ^bc^	725.88 ^a^	132.360	<0.001
*Strecker Aldehydes*							
Acetaldehyde	86.20	116.84	41.46	73.27	58.09	27.305	0.169
Benzaldehyde	36.67 ^c^	23.65 ^c^	62.07 ^bc^	164.42 ^a^	89.35 ^b^	17.477	<0.001
Butyraldehyde	6.06 ^b^	6.49 ^b^	6.36 ^b^	7.41 ^a^	6.39 ^b^	0.309	0.013
Methional	2.28 ^b^	4.66 ^b^	8.38 ^a^	4.89 ^b^	3.77 ^b^	1.088	0.001
Phenylacetaldehyde	8.54 ^bc^	11.85 ^b^	6.10 ^c^	18.21 ^a^	10.10 ^bc^	1.803	<0.001
2-Methylbutanal	18.10 ^b^	23.51 ^a^	12.74 ^c^	20.74 ^ab^	15.56 ^bc^	1.915	<0.001
3-Methylbutanal	15.69	16.91	16.61	16.22	16.26	1.071	0.918
*Sulfides*							
Carbon disulfide	53.96 ^ab^	32.46 ^bc^	15.59 ^c^	74.23 ^a^	20.58 ^c^	11.511	<0.001
Diallyl sulfide	4.21 ^b^	2.91 ^b^	5.36 ^b^	13.61 ^a^	16.63 ^a^	2.249	<0.001
Dimethyl disulfide	0.12 ^b^	0.12 ^b^	0.16 ^b^	0.39 ^a^	0.27 ^ab^	0.058	0.003
Dimethyl sulfide	11.74	10.98	13.46	10.21	8.80	2.020	0.535
Furfuryl sulfide	21.68	20.66	20.85	21.76	27.30	2.962	0.158
*Thiols*							
Methanethiol	25.27	54.48	19.17	28.82	30.68	10.095	0.082
*Thiophenes*							
3-Methyl thiophene	2.47 ^b^	2.44 ^b^	2.46 ^b^	4.08 ^a^	2.46 ^b^	0.272	<0.001

^1^ Includes 15% and 20% fat ground beef. ^2^ Includes less than or equal to 7% fat ground beef. ^3^ Largest standard error of the least squares means. ^4^ Observed significance level. ^a–d^ Means without a common superscript differ (*p* < 0.05).

**Table 4 molecules-28-03151-t004:** Least squares means of volatile compound concentration from three plant-based meat alternatives and two lean levels of ground beef.

	Ground Beef		Plant-Based Meat Alternatives				
Volatile Compound (ng/g)	Lean ^1^	Regular ^2^	Beyond Meat	Impossible Burger	Third Retail Brand	SEM ^3^	*p*-Value ^4^
*Alcohols*							
Ethanol	10.70 ^b^	21.01 ^b^	11.46 ^b^	11.97 ^b^	41.14 ^a^	7.085	0.009
1-Hexanol	3.78	6.06	12.11	4.12	11.42	3.967	0.253
1-Octanol	9.04 ^b^	11.52 ^b^	17.83 ^b^	43.89 ^a^	13.55 ^b^	4.271	<0.001
1-Octen-3-ol	7.19 ^b^	11.32 ^a^	7.11 ^b^	5.88 ^b^	6.10 ^b^	0.930	<0.001
1-Pentanol	5.76 ^b^	9.66 ^a^	5.16 ^b^	5.09 ^b^	6.01 ^b^	0.908	<0.001
1-Penten-3-ol	46.00 ^ab^	65.53 ^a^	19.90 ^c^	54.20 ^ab^	31.82 ^cb^	9.322	0.005
2,3-Butanediol	16.44	19.67	12.41	25.89	13.17	3.997	0.052
*Aldehydes*							
Decanal	9.89	10.82	19.03	20.54	13.11	3.850	0.132
Dodecanal	18.91	16.20	59.17	17.84	59.61	16.756	0.106
Heptanal	9.30 ^c^	20.42 ^ab^	26.36 ^a^	7.53 ^c^	13.50 ^bc^	2.591	<0.001
Hexanal	39.98 ^c^	75.63 ^bc^	197.61 ^a^	103.87 ^b^	136.41 ^ab^	23.278	<0.001
Nonanal	30.23 ^c^	48.12 ^abc^	69.56 ^a^	35.45 ^bc^	56.09 ^ab^	8.879	0.011
Octanal	11.81 ^c^	22.63 ^ab^	26.85 ^a^	14.89 ^bc^	17.31 ^bc^	3.145	0.005
Pentanal	3.85 ^b^	6.89 ^b^	12.90 ^a^	5.53 ^b^	15.49 ^a^	1.775	<0.001
2-Undecenal	16.80 ^b^	26.67 ^b^	34.83 ^b^	16.78 ^b^	81.79 ^a^	7.513	<0.001
(E,E)-2,4-Decadienal	15.57 ^b^	17.22 ^b^	174.19 ^a^	17.73 ^b^	95.07 ^ab^	34.280	0.002
*Carboxylic acids*							
Acetic acid	190.81 ^c^	236.92 ^c^	439.27 ^bc^	668.79 ^ab^	968.49 ^a^	131.410	<0.001
Butanoic acid	9.40 ^ab^	15.43 ^a^	9.37 ^ab^	5.78 ^b^	9.69 ^ab^	2.387	0.049
Heptanoic acid	14.77	14.74	13.88	14.14	15.10	0.420	0.207
Nonanoic acid	8.38	6.51	7.65	6.73	7.52	1.117	0.755
Octanoic acid	76.96 ^b^	48.99 ^b^	124.33 ^b^	65.14 ^b^	396.33 ^a^	45.120	<0.001
*Esters*							
Methyl butyrate	4.56 ^b^	6.26 ^b^	5.07 ^b^	157.49 ^a^	23.02 ^b^	34.893	<0.001
Methyl hexanoate	1.53 ^ab^	1.82 ^a^	1.27 ^ab^	1.12 ^b^	1.31 ^ab^	0.234	0.276
Methyl nonanoate	5.71 ^b^	5.73 ^b^	5.64 ^b^	5.71 ^b^	8.69 ^a^	0.567	<0.001
Methyl octanoate	2.79 ^a^	2.88 ^bc^	3.10 ^ab^	2.84 ^bc^	3.40 ^a^	0.118	<0.001
Methyl propionate	10.47 ^abc^	14.88 ^a^	4.09 ^c^	11.72 ^ab^	5.48 ^bc^	2.706	0.026
*Furans*							
2-Pentyl furan	3.24 ^b^	4.04 ^b^	24.17 ^a^	6.85 ^b^	25.46 ^a^	3.195	<0.001
*Hydrocarbons*							
Benzene	0.18	0.21	0.36	0.22	0.25	0.059	0.244
Decane	4.84	7.94	6.28	4.47	7.51	1.438	0.303
Ethylbenzene	2.80 ^ab^	2.65 ^bc^	3.12 ^a^	2.33 ^c^	2.41 ^c^	0.144	<0.001
Nonane	3.44 ^bc^	6.35 ^b^	13.21 ^a^	2.99 ^c^	3.93 ^b^	1.181	<0.001
Octane	7.04 ^b^	15.81 ^a^	20.80 ^a^	4.59 ^b^	4.94 ^b^	2.429	<0.001
Styrene	2.86 ^ab^	2.70 ^bc^	3.20 ^a^	2.38 ^c^	2.46 ^c^	0.149	<0.001
Tetradecane	2.17 ^a^	2.66 ^ab^	3.71 ^a^	1.81 ^b^	4.64 ^a^	1.111	0.031
Toluene	4.37	4.89	4.67	4.22	3.81	0.456	0.364
*p*-Xylene	55.95 ^b^	53.18 ^b^	246.45 ^a^	200.66 ^a^	28.51 ^b^	29.375	<0.001
1-Octene	2.37 ^b^	5.60 ^a^	2.03 ^b^	1.23 ^b^	2.76 ^b^	0.688	<0.001
*Ketones*							
2-Butanone	14.32 ^b^	19.34 ^b^	17.75 ^b^	30.33 ^a^	17.91 ^b^	3.589	0.012
2-Heptanone	4.08 ^c^	6.76 ^c^	21.96 ^a^	4.70 ^c^	14.21 ^b^	2.019	<0.001
2-Pentanone	0.27	0.32	0.27	0.30	0.25	0.035	0.627
2-Propanone	129.65 ^b^	169.33 ^b^	200.67 ^b^	395.73 ^a^	207.80 ^b^	39.614	<0.001
*Lactones*							
Butyrolactone	15.54	15.97	13.37	12.77	12.70	4.664	0.904
*Terpenes*							
α-Pinene	2.57 ^b^	2.53 ^b^	2.45 ^b^	2.38 ^b^	2.89 ^a^	0.131	<0.001
d-Limonene	3.35 ^b^	3.44 ^b^	4.61 ^b^	2.92 ^b^	14.96 ^a^	1.376	<0.001

^1^ Includes 15% and 20% fat ground beef. ^2^ Includes greater than or equal to 7% fat ground beef. ^3^ Largest standard error of the least squares means. ^4^ Observed significance level. ^a–c^ Means without a common superscript differ (*p* < 0.05).

## Data Availability

Not applicable.

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
