# Peer review of "Descriptive Sensory Attributes and Volatile Flavor Compounds of Plant-Based Meat Alternatives and Ground Beef"

_molecules, 2023, doi:10.3390/molecules28073151_

Round 1

Reviewer 1 Report

The topic of this article is interesting because it is well known that consumers are becoming more interested on opting to replace meat in their diet or decrease their meat consumption and search for plant-based meat alternatives. Therefore, the goal of this study to characterize the differences in sensory and volatile flavor profile between ground beef (GB) and plant-based meat alternatives (PBMA) available at retail in the United States is exciting. The statistical treatment of results is adequate. Results are clear and the discussion is well done. However, abstract and introduction sections are very poor and need improvement.

 The abstract needs to be completely re-written, it should a brief description the methods used for volatile compounds analyses and descriptive sensory analysis, and a brief description of major conclusions, globally and not detail specific aspects. “Patties were evaluated for 12 volatile compounds analyzed by headspace via solid phase  microextraction coupled to GC-MS. Descriptive sensory analysis used a trained panel for flavor and texture attributes. Statistical treatment of results was done by…

Concerning the conclusions, which is the key contribution of this work? The novelty of this study should be highlighted

It should be clear in section 4.1. which are plant-based meat alternatives (PBMA)? Beyond  Burger (BEY); Impossible Burger (IMP); a third brand of PBMA (GEN), and which are the samples of ground beef? regular (RGB); lean GB 10 (LGB), I suppose.  No information was provided in the abstract concerning BEY and IMP, so it is not needed to detail their coded on the abstract.

Author Response

The authors would like to thank Reviewer 1 for their feedback on our submitted manuscript “Descriptive sensory attributes and volatile flavor compounds of plant-based meat alternatives and ground beef”. The abstract was rewritten to fit the model of Molecules. Additional information was added to the introduction to increase background knowledge.

Reviewer 2 Report

This study investigated the different flavor attributes and volatile flavor profile between ground beef (GB) and plant-based meat alternatives (PBMA). It is interesting and valuable to the journal’s readers. However, the manuscript needs revisions and the followings are some comments and suggestions for authors to consider and improve the manuscript.

1.      Table 1 should be moved into the supplementary materials.

2.      For identification of volatile compounds, the retention time and fragmentation pattern of 3 key ions should be provided and added in the supplementary materials.

3.      The sentence for “This section may be divided by subheadings. It should provide a concise and precise 60 description of the experimental results, their interpretation, as well as the experimental 61 conclusions that can be drawn.” should be deleted.  

4.      The standard deviation (SD) should be provided for all the tables.

5.      The odor activity value (OAV) for the volatile compounds should be calculated and key differential compounds should also be indicated to distinguish the aroma profiles of GB and PBMA.

6.      The odor description of volatiles also should be provide for the table 3 and table 4.

7.      More related references are needed to discuss the results obtained.

Author Response

The authors would like to thank Reviewer 2 for their feedback on our submitted manuscript “Descriptive sensory attributes and volatile flavor compounds of plant-based meat alternatives and ground beef”.

In regard to Comment 1, Table 1 has been moved into supplementary materials. Additionally, a second supplementary table was generated including retention times, fragment ions, and odor descriptors of each volatile compound reported in the manuscript (Comments 2 and 6). The section of highlighted by the reviewer was deleted (Comment 3). We reported the standard error of the mean as the measure of variation (comment 4). We have opted to not include OAV in the paper. There is variation in the odor thresholds reported in the literature, especially when accounting for the matrix in which the odors are measured i.e., oil, water, etc. We have added additional references in the discussion to increase clarity.

Reviewer 3 Report

The article entitled “Descriptive sensory attributes and volatile flavor compounds of plant-based meat alternatives and ground beef”, submitted to the journal presents a detailed comparative study of two types of ground beef burgers, three brands of plant-based proteins and ground beef. The issue of textural and sensory properties when replacing animal products with alternative plant products is gaining importance in connection with climate changes and a healthy lifestyle.

The manuscript is clearly structured, the range of analyses used is suitable for the declared aim of the study. The results were statistically processed and are mostly satisfactorily discussed.

I have the following remaks on the article:

-          The authors described differences in the volatile and sensory profiles of GB and PBMA. What is their opinion on ways to approach the sensory properties of heat-processed meat products and plant-based alternatives?

-          There are minor errors in the text, e.g. duplicate marking of figures with the number 2.

 On my opinion, the work brought a number of findings for further research and practice and manuscript could be accepted for the publication after minor revision.

Author Response

The authors would like to thank Reviewer 3 for their feedback on our submitted manuscript “Descriptive sensory attributes and volatile flavor compounds of plant-based meat alternatives and ground beef”. We have addressed the discrepancies in table numbers.

"The authors described differences in the volatile and sensory profiles of GB and PBMA. What is their opinion on ways to approach the sensory properties of heat-processed meat products and plant-based alternatives?".

Response: Thank you for your comment. There are various ways to evaluate the sensory properties of heat processed (not fully cooked) ground beef and plant-based meat alternatives. A descriptive odor panel could be used to quantify the odors present in raw product. Moreover, HS-SPME GCMS can be used to evaluate the volatile profile of the raw samples.

Round 2

Reviewer 2 Report

The authors revised some of the issues. However, I have not found the supplementary materials. In addition, the odor activity value (OAV) for the volatile compounds should be calculated and key differential compounds should also be indicated to distinguish the aroma profiles of GB and PBMA. The authors have not addressed this issue.

Author Response

Thank you for your response. The supplementary materials are presented in Tables at the end of the Manuscript draft. Supplementary table 1 includes the sensory lexicon and supplementary table 2 includes the volatile compound information. In regard to OAV, we did respond indicating we chose not to include those calculations. Beef and PBMA matrices are complex and published aroma thresholds are variable for a single compound.